# The role of Zambia's expansive Inter-agency Coordinating Committee (ICC) in supporting evidence-based vaccine and health sector programming

Zoe Sakas[1], Katie Rodriguez[1], Kyra A. Hester[1], Roopa Darwar[1], Bonheur Dounebaine[1], Anna S. Ellis[1], Simone Rosenblum[2], Kimberley R. Isett[3], Walter Orenstein[4], Matthew C. Freeman[1], William Kilembe[5], Robert A. Bednarczyk[6]*

1 Gangarosa Department of Environmental Health, Rollins School of Public Health, Emory University, Atlanta, Georgia, United States of America, 2 School of Public Policy, Georgia Institute of Technology, Atlanta, Georgia, United States of America, 3 Biden School of Public Policy and Administration, University of Delaware, Newark, Delaware, United States of America, 4 Emory School of Medicine, Emory University, Atlanta, Georgia, United States of America, 5 Center for Family Health Research in Zambia, Lusaka, Zambia, 6 Hubert Department of Global Health, Rollins School of Public Health, Emory University, Atlanta, Georgia, United States of America

* robert.a.bednarczyk@emory.edu

**Data Availability Statement:** Requests for deidentified data used to support this manuscript can be made to Marina Bruck,  marina.

## Abstract

New vaccines, technologies, and regulations, alongside increased demand for vaccines, all require prioritization and coordination from key players within the vaccine sector. Inter-agency Coordinating Committees (ICC) support decision-making and coordination at the national-level and act as key drivers for sustainable improvements in vaccination programming. We utilized a previous qualitative case study, which investigated critical success factors for high routine immunization coverage in Zambia from 2000–2018, specifically to study the Zambian ICC. Qualitative data were collected between October 2019 and February 2020, including key informant interviews (n = 66) at the national, provincial, district, and health facility levels. Thematic analysis was applied to understand the role of the Zambian ICC and its impact on the policy environment over time. Within our study period, the ICC demonstrated the following improvements: 1) expanded membership to include diverse representation; 2) expanded scope and mandate to include maternal and child health in decision-making; and 3) distinct roles for collaboration with the Zambia Immunization Technical Advisory Group (ZITAG). The diverse and expansive membership of the Zambian ICC, along with its ability to foster government commitment and lobby for additional resources, supported improvements in immunization programming. The Zambian ICC holds considerable influence on government agencies and external partners, which facilitates procurement of funding, policy decisions, and strategic planning.

bruck@emory.edu, a project coordinator managing infectious disease-related data at Emory University Rollins School of Public Health. Review of these requests to maintain compliance with ethical reviews guiding this project will also need to be communicated to the Emory University IRB at irb@emory.edu.

**Funding:** This work was supported by the Bill & Melinda Gates Foundation, grant number OPP1195041 (awarded to MCF); salary support was provided to ZS, KR, KAH, RD, BD, ASE, KRI, WO, MCF, RAB, and a subaward was granted to WK. Pilot and proposal development funds were provided by Gates Ventures (MCF); salary support provided to KH, RAB, MCF. The Bill & Melinda Gates Foundation and Gates Ventures worked with the study team to provide technical assistance and assembled the Research Advisory Group. All other aspects—i.e., study design, materials, methodology, data collection and analysis, and manuscript writing—were done solely by the study team.

**Competing interests:** The authors have declared that no competing interests exist.

# 1. Introduction

Immunization policies and programs in low and middle-income countries are constantly evolving [1]. The introduction of new vaccines, technologies, and regulations—alongside a general global increase in demand—all require prioritization and coordination from key players within the vaccine sector [1]. To respond to growing needs, the establishment and maintenance of effective, efficient, and rigorous national-level coordination forums are essential.

In the 1990s, the World Health Organization (WHO) established Inter-Agency Coordinating Committees (ICCs) as country-level forums to coordinate polio eradication programming [2]. In 2006, Gavi, the Vaccine Alliance (Gavi) began using ICCs to support broader management of funds for immunization programming [3, 4]. Gavi also required creation of National Immunization Technical Advisory Groups (NITAGs) to present data and evidence-based recommendations to the ICC and other national forums [5–7]. Since 2001, Gavi has contributed $164,000,000 to the Zambian government for vaccination programming and health systems strengthening, which has supported the expansion and strengthening of Zambia's ICC and the Zambian Immunization Technical Advisory Group (ZITAG) [7, 8].

Literature suggests that ICCs and NITAGs should have distinct roles–the ICC supports supervision and coordination, and the NITAG provides technical expertise to promote rigorous decision-making [1, 7]. However, relying on parallel administrative structures may foster confusion around roles and responsibilities, leading to inefficiencies and disruptions in program functioning [6, 9, 10]. Tailoring the roles and responsibilities of these unique forums to ensure they complement, rather than disrupt, each other may help overcome these challenges.

Successful coordination forums act as key drivers for sustainable improvements in vaccination programming and coverage [1, 5, 11]. This study illustrates how Zambia bolstered its ICC for long-term functionality, expanded ICC membership and scope beyond Gavi requirements, and distinguished a complementary structure for the ZITAG. Zambian stakeholders reported that this approach worked well to support immunization programming and improve vaccination coverage [11].

# 2. Methods

This study is nested within a larger research project, Exemplars in Vaccine Delivery, which investigated key success factors for improvements in routine immunization coverage in Zambia, Nepal, and Senegal. Zambia was chosen for this case-study as the ICC, and the policy environment in general, emerged as a more significant theme in the qualitative data as compared to the other exemplar countries. While each country has a working ICC, key informants in Zambia spoke about the ICC and ZITAG frequently when asked about success factors that contributed to vaccination coverage. The emergence of the ICC as a unique factor prompted this study as to allow for a more thorough analysis of this topic.

## 2.1 Research questions

1. What were key drivers of success for high and sustained routine immunization coverage?

2. What were the key implementation and change management strategies employed for high growth in vaccine coverage levels?

3. How has the policy and decision-making environment changed over time, and how have these changes (or lack thereof) impacted both the functioning of the vaccination program and the environment in which it works?

4. How are country wide policy decisions made?

5. What is the interplay of the NITAG and ICC, and how does this top-down leadership impact decision-making?

This paper details one of the most influential success factors described by key informants–the role of the Zambian ICC–and its impact on the policy environment from 2000–2018.

## 2.2 Study design and setting

We employed a qualitative case study design to investigate factors that supported improvements in routine immunization coverage for children under 1 year of age in Zambia from 2000 to 2018. Zambia was one of three countries selected as exemplars in vaccine delivery [12]. Country selection for exemplars was determined by high and sustained coverage of the first and third doses of the diphtheria-tetanus-pertussis vaccine (DTP1 and DTP3; illustrated in Figs 1 and 2, respectively), which served as proxies of the vaccine delivery system [13, 14]. Lusaka, Central, and Luapula provinces were selected based on vaccination coverage and contextual factors. Fig 3 illustrates routine immunization coverage improvements in different regions in Zambia to demonstrate heterogeneity between study sites [11, 14]. Additional information about the site selection, sampling, and protocol can be found elsewhere [11, 12].

## 2.3 Data collection and analysis

Qualitative data were collected between October 28, 2019, and March 6, 2020. Data collection included key informant interviews (n = 66) at the national, provincial, district, and health facility levels. Data collection activities are summarized in Table 1.

Data were collected by the Center for Family Health Research in Zambia (CFHRZ). Key informant interview (KII) guides were translated into Nyanja and Bemba languages by research assistants. Topic guides aimed to identify key factors that drove the success of the vaccination program in Zambia, especially during points of catalytic growth in DTP1 and DTP3 coverage from 2000–2018. All interview guides were piloted before use and adjusted iteratively throughout data collection.

An initial list of KIIs was developed with CFHRZ and MoH officials; we then used snowball sampling to identify additional key informants. The duration of KIIs averaged one and a half hours. KIIs were audio-recorded with the permission of participants. Research files,

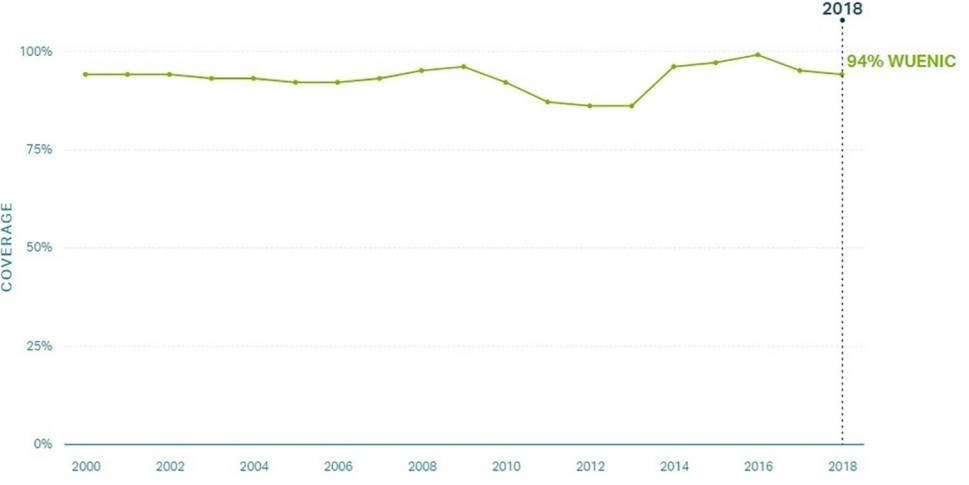

**Fig 1. DTP1 coverage in Zambia 2000–2018 [13].**

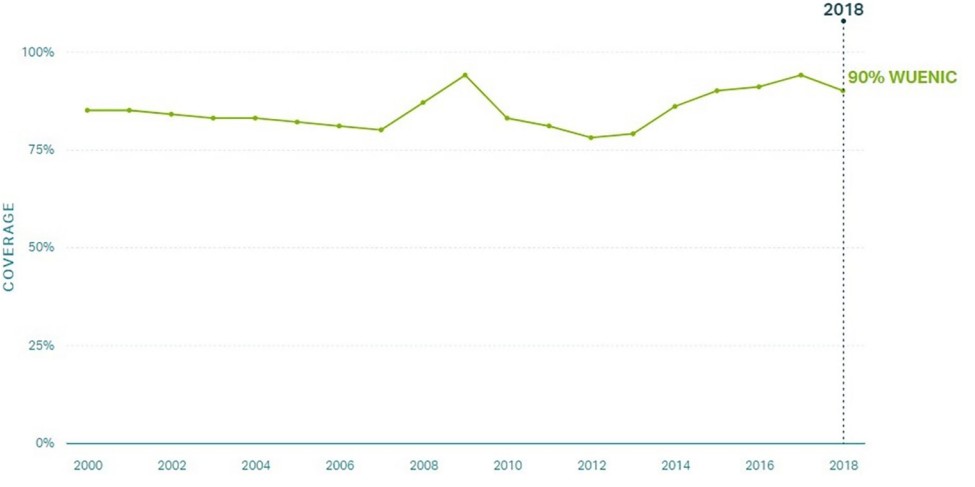

**Fig 2. DTP3 coverage in Zambia 2000–2018 [14].**

recordings, and transcriptions were de-identified and password protected. Debriefs were conducted during data collection to identify emerging themes and follow-up questions. All the topic guides used for this study, as well as our analysis tools, can be found on our Open Sciences Framework webpage [15].

We conducted a thematic analysis of the transcripts to identify contributing factors to the success of the immunization program. We developed a codebook using a deductive approach, applying constructs from existing implementation science frameworks—notably the Consolidated Framework for Implementation Research (CFIR) and the Context and Implementation of Complex Interventions (CICI) frameworks [16, 17]. The codebook was also iteratively adjusted based on emerging themes. All transcripts were coded and analysed using

## ZAMBIA IMPROVED DTP3 COVERAGE THROUGHOUT ALL REGIONS

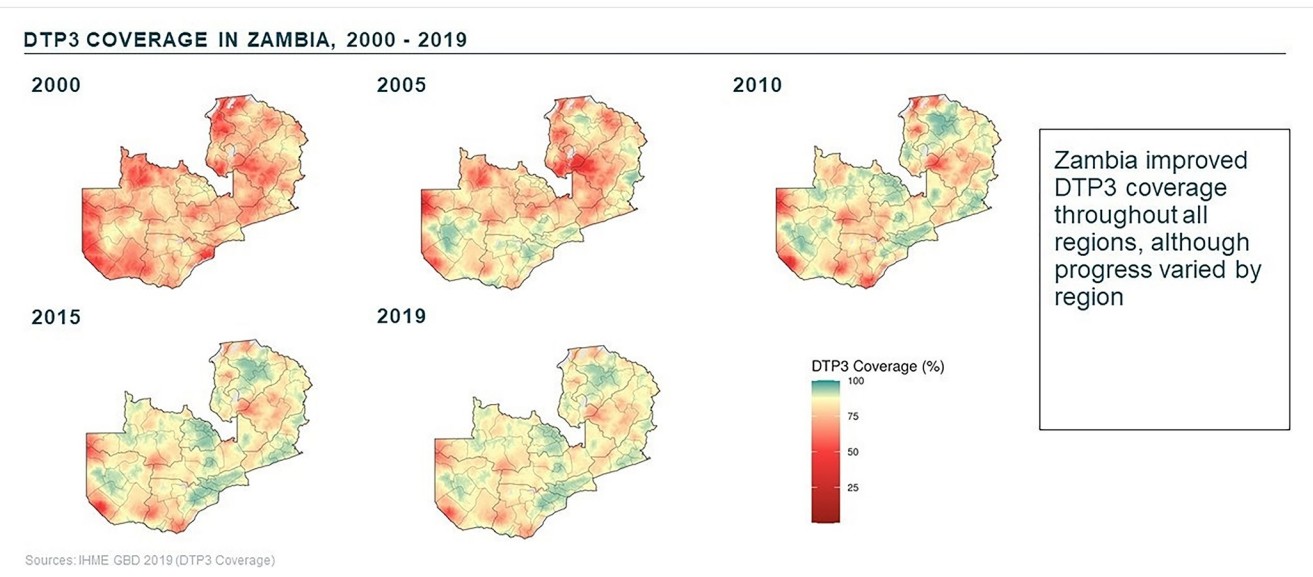

**Fig 3. Heat map illustrating improved DTP3 coverage across all regions of Zambia 2000–2019 [14].**

**Table 1. Summary of research activities, October 2019 –February 2020.**

| Method | Participants | Number of activities | Number of participants |
|---|---|---|---|
| Key Informant Interviews | Ministry of Health | 9 | 10 |
| | Ministry of Education | 1 | 1 |
| | Ministry of Finance | 1 | 1 |
| | Partner organization | 11 | 15 |
| | Provincial Health Office | 6 | 8 |
| | District Health Offices | 10 | 19 |
| | Nurses in-charge of health facilities | 7 | 10 |
| | Community-based volunteers | 11 | 11 |
| | Community leaders | 10 | 10 |
| **Total** | | **66** | **85** |

MaxQDA2020 software (Berlin, Germany). To ensure consistency between coders, qualitative analysts performed regular intercoder reliability tests and attended frequent coding meetings to address discrepancies, discuss emerging themes, and add empirical codes to the codebook. Relevant themes were identified through our frameworks alongside deductive reasoning, and visual tools were used to illustrate the findings. We considered setting and participant roles while identifying key points, and further contextualized data using historical documents and a literature review. We used key words (e.g., "Zambian ICC", "ZITAG", "Expanded Programme on Immunization", "vaccination in Zambia") to search Zambian government websites and academic journals. Further information about data management and analysis can be found in our previously published work [11, 12].

## 2.4 Gavi document review

Following qualitative data analysis, we reviewed all publicly available Gavi annual reports for Zambia within our study period—including joint appraisals [5]. During this document review, we focused on adding context and information to the themes that emerged from the qualitative data. Findings from the qualitative data were triangulated with information from Gavi documents to strengthen the analysis.

## 2.5 Research advisory group and stakeholder feedback

Findings were shared with a research advisory group composed of international and Zambian stakeholders, which included experts in the governance, policy, and immunization sectors. Follow-up questions were shared with in-country key informants to gain more information and detail. Details about advisory group members can be found in our published protocol [12].

## 2.6 Ethics statement

The study was approved by the University of Zambia Biomedical Research Ethics Committee (Federal Assurance No. FWA00000338, REF. No. 166–2019), the National Health Research Authority in Zambia, and the Institutional Review Board committee of Emory University, Atlanta, Georgia, USA (IRB00111474). Informed consent was obtained from all 85 participants. Written consent forms were provided, reviewed, and signed prior to interviews. All consent forms were approved by the appropriate ethics committees.

## 3. Results

According to our analysis, the Zambian ICC has held considerable influence on government agencies and external partners, which facilitates the procurement of funding, policy decisions, and strategic planning.

> "*For the EPI [Expanded Programme on Immunization], the Inter-Agency Coordinating Committee is a very important organ for policy development and resource mobilization.*" *(WHO personnel)*

The Zambian ICC was established in 1999 to meet WHO Polio Eradication Initiative requirements. In 2006, the ICC was adapted to support the management of Gavi funding for general immunization programming. The ICC continues to oversee vaccine initiatives, lobby for funding from the national government and partners, and foster collaboration between the Ministry of Health (MoH), partners, and community-level stakeholders [5]. The key responsibility of the ICC is to facilitate decision-making and resource mobilization through monthly meetings.

Throughout our study period, particularly from 2014 to present, the Zambian ICC has demonstrated the following improvements: 1) expanded membership to include diverse representation; 2) expanded scope and mandate to include maternal and child health in decision-making; and 3) collaboration with the ZITAG through distinct and complimentary roles and responsibilities. These factors are described further below.

### 3.1 The Zambian ICC expanded its membership and scope to improve reach, coordination, and commitment

According to key informants and Gavi joint appraisals from 2014 to 2018, the Zambian ICC performed above the minimal requirements obligatory for funding [5]. The ICC sufficiently accomplished recommendations from Gavi related to strategic planning, financing, membership, information dissemination, and decision-making procedures which are outlined in joint appraisal reports [5]. These improvements were likely fuelled by recommendations from meetings in 2016 and 2017 which included Gavi representatives, current ICC members, and key stakeholders from the Zambian government. The extent that the ICC improved from 2014 to 2018 suggests commitment and motivation from internal leadership and champions. Specifically, the Zambian ICC expanded both its membership and mandate to promote decision-making and resource allocation.

**3.1.1 Diverse and representative ICC membership.** The Zambian ICC includes diverse and representative membership to support collaboration among technical experts, decision-makers, and community organizations [7]. With its multi-sectoral composition, the ICC provides a forum for coordination of immunization investments, bolsters management of key action points, and administers technical working groups [18, 19]. The MoH holds a leadership position and is considered a "*driving force*" of the forum's success, according to national-level stakeholders. Additionally, over 50% of Zambia's ICC members report having been members of the ICC for over 6 years, according to a recent Gavi evaluation [7]. Diversity and longevity of ICC members was described as beneficial to decision-making.

All members of the ICC are expected to share their opinions and vote on decisions to support collaboration between entities represented on the forum [5, 18, 19]. Additionally, members of the ICC may participate in other forums that contribute to development in Zambia, including meetings with the MoH to develop national health strategic plans and engagement with advisory committees for other health sectors [7]. This cross-membership supports

efficiency of planning and resource allocation for health and development in Zambia through the alignment of ideas and priorities between decision-making bodies. Existing literature states that members of the ICC, ZITAG, and EPI Technical Working Groups are generally informed about decisions, plans, and evaluations in Zambia—which aligns with our findings [7].

Table 2 outlines key ICC membership required or recommended by Gavi and compares the fulfilment of these guidelines by the Zambian ICC in 2014 and 2018 [5].

Gavi support is reliant on maintaining an active ICC, and requires at least one member from each of the following categories:

1. Ministry of Health,

2. EPI Programming,

3. Financial Planning,

4. Health Systems Strengthening,

5. Key Donors, and

6. Implementing Partners.

   Gavi recommends ICC representation from the following categories:

6. Immunization Experts,

8. Technical Advisory Groups, and

9. Private Sector.

In 2018, Zambian ICC members filled all membership categories. The drastic change in ICC membership from 2014 to 2018, as illustrated by Table 2, demonstrates Zambian commitment to strengthening the ICC and improving collaboration for immunization programming. Although membership expanded significantly over the decade, there are still apparent gaps, including limited private sector representation. Additionally, recent reports highlight concerns that some members were in diplomatic positions within their organizations and might be lacking the required technical capacity [7].

The expansive membership of the ICC affected various components of vaccination programming, including cold chain expansion and data quality, through the inclusion of representatives from sub-committees. The EPI in Zambia includes a technical working group, which functions separately from the ICC. The EPI technical working group (EPI-TWG) is divided into four sub-committees: 1) Monitoring and evaluation, 2) Social mobilization, 3) Cold chain and logistics, and 4) Service delivery. EPI-TWG subcommittees are highly active and substantially contributed to the success of the Zambian ICC—particularly by providing reports and evidence to both the ICC and ZITAG [7]. A representative for each sub-committee is represented on the ICC and attends quarterly meetings for decision-making and resource mobilization. Additionally, integrated meetings, attended by ICC members and representatives from the national, provincial, and district levels, are conducted to review data and address gaps in coverage.

Membership of the ICC includes, as illustrated in Fig 4:

- External donors and implementing partners to align priorities and foster mutual trust.

- EPI officers, including representatives from all districts and sub-committees to present sector-specific updates and support context specific implementation.

- Select representatives from technical institutions, including researchers from the Centre for Infectious Disease Research in Zambia (CIDRZ), to ensure evidence is considered.

**Table 2. ICC membership matrix from 2014 and 2018\*.**

| Category | Member | 2014 | 2018 |
|---|---|---|---|
| Ministry of Health<br>*Required* | Minister of Health | X | X |
| | Permanent Secretary | | X |
| | Department of Child Health | | X |
| | Department of Epidemiology | | X |
| | Department of Planning | | X |
| | Department of Health Promotion | | X |
| | Statistical Office | | X |
| Expanded Programme for Immunization (EPI)<br>*Required* | EPI Manager | | X |
| | EPI Officers: Cold chain, M&E | | X |
| | EPI district representatives | | X |
| Financial Planning<br>*Required* | UN Children's Emergency Fund (UNICEF) | X | |
| | World Health Organization | X | X |
| | Gavi, the Vaccine Alliance | X | X |
| | Japan International Cooperation Agency | X | |
| | Zambian Department of Planning | | X |
| | UN Development Programme | | X |
| Health Systems Strengthening<br>*Required* | Minister of Health | X | |
| | Permanent Secretary | | X |
| | UN Development Programme | X | |
| | World Health Organization | X | X |
| | UN Children's Emergency Fund (UNICEF) | | X |
| | UN Population Fund (UNFPA) | | X |
| | U.S. Centers for Disease Control and Prevention | | X |
| Key Donors<br>*Required* | Gavi, the Vaccine Alliance | X | X |
| | Japan International Cooperation Agency | X | |
| | World Bank | | X |
| | UK Department for International Development (DFID) | | X |
| | U.S. Agency for International Development (USAID) | | X |
| Implementing Partners<br>*Required* | Churches Health Association of Zambia (CHAZ) | X | X |
| | World Health Organization | X | X |
| | PATH and Better Immunization Data (BID) Initiative | X | X |
| | UN Children's Emergency Fund (UNICEF) | X | X |
| | World Vision | X | |
| | Catholic Relief Services | X | |
| | ZAMRA | | X |
| | John Snow International | | X |
| | Centre for Infectious Disease Research in Zambia (CIDRZ) | | X |
| | Swedish International Dev't Cooperation Agency | | X |
| | United Nations Development Programme | | X |
| | Centers for Disease Control and Prevention (CDC) | | X |
| Experts<br>*Recommended* | Centre for Infectious Disease Research in Zambia (CIDRZ) | X | X |
| | Zambia School of Nursing | | X |
| | Chimana College | | X |
| | Medical University | | X |

*(Continued)*

**Table 2.** (Continued)

| Category | Member | 2014 | 2018 |
|---|---|---|---|
| Technical Advisory Groups Recommended | Child Health TAG | X | |
| | RMNCAH-N ICC | X | X |
| | Centre for Infectious Disease Research in Zambia (CIDRZ) | | X |
| | Churches Health Association of Zambia (CHAZ) | | X |
| | PATH | | X |
| | Zambia School of Nursing | | X |
| Private Sector *Recommended* | John Snow International | | X |

*Information sourced from Gavi joint appraisals and key informant interviews

*"The ICC is used for [government officials] or partners who support programs in Zambia. It is a committee for high profile [officials] and those who have interest in immunization. Together, we tell them to [describe] the performance of the program and some of the challenges that it is facing, so that people can lobby for support in terms of finances and any other things that we have had tried from the program operation." (EPI Cold Chain Officer)*

**3.1.2 Expanded mandate to other areas of health.** In 2017, the terms of reference for the Zambian ICC expanded to include a broader focus on Reproductive, Maternal, Neonatal, Child, and Adolescent Health, and Nutrition (RMNCAH-N). This decision, discussed during an annual appraisal meeting with key stakeholders, stemmed from a suggestion by Gavi to the Zambian ICC to expand its scope [5]. Following the appraisal meeting, the ICC mandate expanded to include organizations that work in maternal and child health. Key informants from the national and regional level spoke about the positive impact of this expanded mandate, which allows for efficient and comprehensive strategic planning and resource allocation.

*"In an effort to strengthen the governance and oversight capabilities of the ICC, there are recommendations to prioritize ICC strengthening in 2018 through an assessment which will include a comprehensive review of both the [terms of reference] and membership of the group." (Gavi appraisal, 2017)*

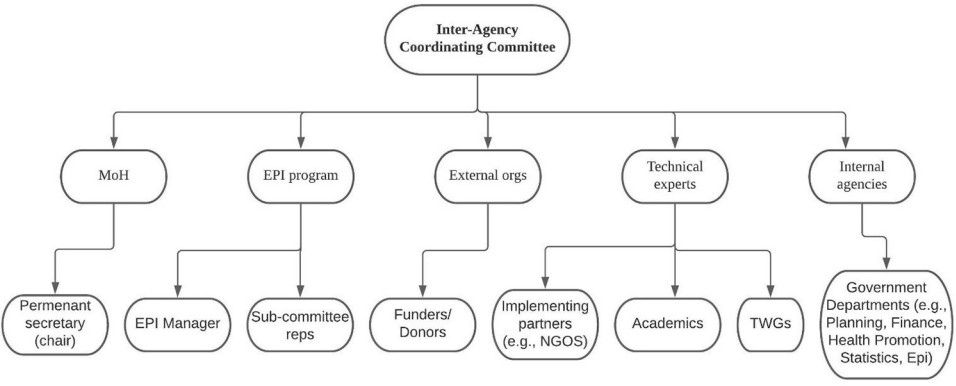

**Fig 4. Description of the ICC structure in Zambia.**

*"The ICC is ideally targeted at the heads of organizations who have an interest in reproductive and maternal and child health and nutrition issues in the country." (UNICEF personnel)*

Although Gavi appraisals stated that the Zambian ICC began to incorporate broader areas of health in 2017, there is evidence from interviews that the ICC was working to expand its scope to include maternal, new-born, and child health representation as early as 2010. This expansion allowed for a more holistic approach to vaccination, involving ante-natal clinics, traditional birth attendants, nurses, and other staff members from health facilities. Activities such as child growth tracking and school enrolment were also tied to vaccination efforts.

*"In terms of decision-making at a higher level, the ICC evolved some time back. There was the RMNCAH-N partnership for maternal, newborn, and child health. It must have been 2010, there was a partnership which was created to incorporate the maternal component." (WHO personnel)*

The ICC's motivation to include RMNCAH-N was likely in response to prioritization of women, early childhood, and adolescent health within the global health policy environment; however, this motivation was not entirely clear from our investigation. It is possible that introduction of the Human Papillomavirus (HPV) vaccine (piloted and scaled up in Zambia in 2013 and 2019, respectively) may have supported this expansion. Representatives focused on HPV immunization were likely involved with the ICC at that time, establishing a natural connection between reproductive and vaccination programming, highlighting the importance of integrated programming and budgeting [20].

### 3.2 Distinct and complementary roles of the Zambian ICC and ZITAG

Our data illustrates that strengthening the relationship between the Zambian ICC and ZITAG for long-term functionality supported improvements in vaccine programming. However, other recent studies highlight the challenges of coordinating between two similar national forums [7]. The ZITAG was established in 2016 to provide evidence-based recommendations and as a requirement to continue receiving Gavi funding [7, 8]. To date, the ICC and ZITAG have distinct and complementary roles and responsibilities, as described in Fig 5 and Table 3. Gavi-funded programs typically focus first on the adoption of an ICC to manage donor funding, and later, the establishment of the NITAG as a parallel technical group. Key informants from the MoH described that the collaboration between the Zambian ICC and ZITAG allowed for long-term functionality without duplication of efforts.

Through formal interactions, the ZITAG empowers the ICC to make evidence-based decisions. The ZITAG presents evidence to the ICC either recommending or discouraging a particular strategy or intervention based on context, formative research, piloting, and situational analyses. The MoH considers the evidence presented by the ZITAG when formulating national immunization policies and strategies. Integration of both bodies within national decision-

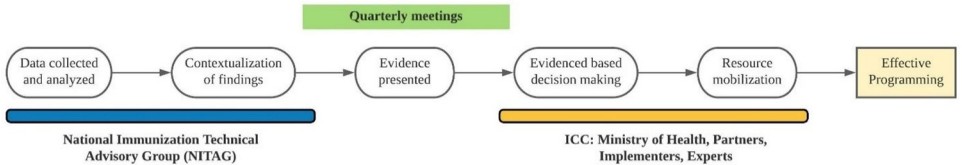

**Fig 5. Programming accomplished through collaboration of the ICC and NITAG in Zambia.**

**Table 3. Comparing mandate and membership of ICCs and NITAGs.**

|  | ICC | NITAG |
|---|---|---|
| *Mandate* | • Fosters collaboration between entities<br>• Supports EPI resource mobilization<br>• Supports advocacy efforts<br>• Acts as a decision-making forum<br>• Holds quarterly meetings<br>• Lobbies for funding | • Collects and analyses data to provide the MoH and ICC with evidence-based advice<br>• Holds an advisory role<br>• Develops and supports grant proposals<br>• Provides technical advice<br>• Supports contextualization |
| *Membership* | • MoH (chair)<br>• Representatives from technical groups<br>• Donors and implementing partners<br>• Civil Society Organizations<br>• EPI officers (including sub-committee members and district reps),<br>• Private sector | • Core members include experts in public health, health care, epidemiology, vaccine sector, sociology, economics<br>• Liaison with technical partners, such as WHO, UNICEF<br>• Secretariat; ensures NITAG is functioning |

making is enhanced by the distinct roles of each entity, a component that is necessary for the success of advisory and coordination committees [1, 6]. Despite its recent establishment, Zambian officials interviewed for this study value the ZITAG's ability to provide context-specific recommendations for immunization programming [21].

> *"In terms of professional bodies in health, we have a good number of them that attend [regular coordination] meetings; also, we have the NITAG that has people pulled from different professions. So that's quite a key component in the governance system." (EPI personnel)*

Additionally, the ZITAG includes several technical stakeholders, such as academics, epidemiologists, health care professionals, scientific societies, and technical experts from NGOs, whereas ICC membership includes a broader range of leadership [21]. Key informants in Zambia agreed that delays from coordination between the ICC and ZITAG are offset by the benefits of more robust, evidence-based decisions that are tailored to local context, which aligns with findings from existing literature [7]. In some cases, members of the ZITAG and EPI-TWG report questioning the efficiency and transparency of the ICC, which suggests that even further improvements in coordination and communication between these entities may be needed.

The addition of a fully functioning ZITAG, alongside the decision-making authority of the ICC, supports strategic planning and advocacy for funding:

> *"The ZITAG is looking at [immunization] from a very critical standpoint, and from a very technical standpoint—looking at costs, disease burden, etc. I think [the ZITAG is] an external body that's incredibly important to give recommendations compared to the ICC which [does not always include] many technical partners."* (CIDRZ personnel)

### 3.3 Additional informative quotes

By speaking with 85 participants from all levels of the Zambian government and throughout the immunisation system, we have a comprehensive collection of illustrative quotes that help contextualize the key findings outlined above and demonstrate the comprehensive nature of this study. Please find a small selection of these quotes in Table 4.

## 4. Discussion

The importance of the Zambian ICC was a reoccurring theme among our key informants—particularly those at the national and regional levels. This topic emerged quickly and stayed

**Table 4. Quotes illustrating the impact of the Zambian ICC on vaccination programming.**

| Topic | Quote | Participant |
|---|---|---|
| Collaboration between the ICC, subcommittees, and technical working groups | "Well, the EPI technical working group is really a subcommittee of the ICC. We have the ICC which goes beyond immunization to include reproductive health, as well as nutrition and other child health interventions or programs. So, the EPI sits under the ICC, and we do have a technical working group which sort of sits more at a technical level and gives information to the ICC for policy direction and decision-making at that level. The technical working groups are smaller subcommittees with specializations. So, there [are] some subcommittees that look at community engagement, there are subcommittees that look at logistics, some that look at service delivery, and so on and [and] so forth. So, the technical working group does have terms of reference, they meet periodically as agreed–generally, they meet at least every two weeks. They can address certain questions or issues and determine how the program can respond to those things and [they] update the ICC on a quarterly basis." | National-level government employee |
| Chain of command in the immunization sector, including the ICC and ZITAG | "We have a very clear chain of command in the sector that we have—on top, the Ministry of Health under the Directorate of Public Health. This is where the EPI or Expanded Program on Immunization sits, and the EPI has a technical working group which brings together the partners, both international and local ones. We also have independent advisors who we meet with when we want to introduce a new vaccine or policy direction that hinges on vaccine safety. We also have the ZITAG. . . that advises the government and partners in the field. We also have the Interagency Coordinating Committee which is the apex of decision makers in terms of programming and policy, and the Minister or the Permanent Secretary is chair of that caucus, and it makes decisions on immunization and other issues including nutrition and MNCH." | CHAZ employee |
| Coordination between different levels of the government and international organizations | "Before 2016, we didn't have the NITAG, so it was [EPI] who presented to senior management, the Ministry of Health. Then senior management would say, 'Oh, let's hear what partners will say,' and then go to the ICC. The ICC would either throw it out or request that we make changes to reflect [their comments], or they adopted it as it is. And then once it is adopted in that request, we would also propose who could potentially fund it, previously we would write a proposal with one funder in mind or mention if we wanted the Ministry of Finance to do this. . . But lately I think most of the proposals have been going through Gavi, so then once ICC says it's okay, it goes right to Gavi. However, there was not a single proposal that we sent to Gavi that was solely funded by them, so it's Gavi, partners, and government, they all have a stake in it. So, once it is approved, then we involve the sub-national levels. When we are doing a proposal, we pick up the phone and call, 'Okay we intend to do to ABCD, how much would it cost?' But I think it is at this stage that they get a lot of involvement—they guide, particularly on the district-level, they guide how many people will be required, what resources will be required, how many vaccines, an estimate of how much money will be required. So, there is that kind of involvement, and then those plans advocated at the national level. And then we get back to ICC [with] a revised budget. Those are the points at which the different levels are getting involved." | EPI manager |
| Funding for immunizations secured by the ICC | "I think it was an initiative from the ICC–of course, we don't want a break in terms of service delivery for immunizations. Another issue is that without such an initiative, we have the risk of procurement of vaccines being a challenge in some areas. You know, the government has a lot of priorities, but this one I think it was started with partners through ICC. . . There should not be any interruption due to shortage of funding, okay. I think it has been catered to make sure that money is separate from the main budget that it is meant for just the supply of immunizations." | MoH officer |
| Inception of the ICC and the important role of Gavi | "Ok, what [happens] is that the program will make recommendations around mainly new vaccine introductions, resource mobilization in terms of budget endorsement, whether or not to implement certain interventions, such as maybe in response to a disease outbreak, whether we have to go the campaign way or find alternative interventions to handle a disease burden. So those are the main areas that the ICC would actually endorse. But some of the decisions were related to Gavi. Gavi requires that for a decision to be tabled before their panel, it must go through the ICC. . . [The ICC became powerful] because of Gavi requirements. However, it is not because Gavi was imposing it on the country, but the country also felt that the intervention would be beneficial to the targeted population. Definitely, the country will go and find evidence that we have a disease burden in this area and if nothing is done this will be the implication, so that which [Gavi] put before us was seen as a potential support, as something that we will benefit from, so we would realign ourselves to meet those conditions. Now, if I recall, what my predecessor did mention to me was that the ICC came in as a consequence of some of Gavi's implementation strategies. To implement the Vaccine Independence Initiative, I think Gavi—as part of initiative- felt that there needed to be put in place a coordinating oversite body." | WHO staff |

*(Continued)*

**Table 4.** (Continued)

| Topic | Quote | Participant |
|---|---|---|
| Inclusive and diverse membership within the ICC | *"[The ICC] includes heads of departments or partners who support the organization programs in Zambia. Among them we have [people] from the WHO, departments from UNICEF, and different organizations like CHAZ, CIRDZ, CARE International, of course, and even JICA, yeah. It is usually a committee for heads of departments–those who have [a common] interest in immunization. [We also have] program officers from the districts, we have cold chain [officers], we have service delivery [officers], we have Monitoring & Evaluation [officers]. These are all part of the ICC...So, in short, what I'm saying is that every program officer for EPI program is part of the ICC."* | Cold chain officer |

relevant throughout the study period, pointing to the importance of the Zambian ICC. Due to the uniqueness of Zambia's ICC, and its emergence as a key success factor for increasing vaccination coverage, we determined there to be utility in a distinct analysis within this environment.

These findings reveal the importance of the structure and role of the Zambian ICC may have utility in providing a rough outline for countries looking to improve immunization coverage. These countries may present a deficit in one of the factors shown to augment the success of the ICC. Countries looking to strengthen their policy environments, streamline decision making, improve communication between national-level stakeholders, or advocate for additional health funding may do so by adapting their ICCs based on the following transferable lessons:

- Expand ICC membership to include a variety of government officials, local representatives, community organizations, external partners, and donors to foster coordination and collaboration across stakeholders.

- Expand ICC mandate and scope to include other areas of health (i.e., reproductive, maternal, and child health) in order to foster integration of programs and health systems strengthening, and decrease siloing of systems, while still prioritizing immunization.

- Establish distinct and complementary roles for the ICC and NITAG to ensure efficient, collaborative, and sustainable integration. Continuously revisit the need for additional communication between groups.

Implementation of transferable lessons from this study should take into consideration local context and potential barriers. For example, key informants interviewed for this study spoke positively about the expanded mandate and cross-sectoral approach, and did not mention any concerns during interviews, while a recent Gavi evaluation revealed that members of the ZITAG and EPI-TWG had some concerns over the recent changes [7, 8]. Namely, ZITAG members reported that the link between them and the Zambian ICC was weakened by the broadening mandate, leading to inefficiencies within the immunization sector; some participants described inefficiencies in decision-making related to vaccine programming stemming from this broad focus [7, 8].

In the 2018 Gavi joint appraisal, the Zambian ICC was described as a *"key driver of sustainable coverage and equity"*, in alignment with our findings [5]. We found that the Zambian ICC collaborated closely with government agencies and external partners, which supported timely procurement of funding, policy decisions, and cohesive strategic planning. Although we did not have data that spoke specifically to the impact of the ICC on equity, our Zambia case study —which includes all key success factors including the ICC—shows improvements in vaccination coverage across all regions, and in both rural and urban areas [11].

There is limited information available comparing the functionality of ICCs in different countries or assessing the interplay between ICCs and NITAGs. From our literature review, we found that Kenya and Tanzania, both with DTP3 coverage comparable to that of Zambia, have also experienced the benefits of expanding their ICC membership [22–26]. The Kenyan ICC also utilizes a multisectoral approach, which includes alignment of priorities with other health sectors, for crafting immunization policy through cooperation with the Kenya NITAG (KENITAG). The ICC of Tanzania schedules additional meetings with its NITAG for strategizing efforts to reduce bottlenecks in delivery and improve uptake of immunization services [26]. While all countries receiving Gavi funding are advised to establish an ICC and NITAG, only some have reported fully utilizing collaboration of the two bodies for decision-making [21, 27]. With active, inclusive, and collaborative ICCs and NITAGs, Zambia, Kenya, and Tanzania utilized these forums through defining clear roles and prioritizing the efficacy of immunization programs.

This study has several limitations. First, our mandate was to utilize a positive deviance lens to study immunization coverage factors in our exemplar countries; however, we were unable to carry out a counterfactual analysis in a non-exemplary country. Second, our research tools focused on the factors that drove catalytic change and did not probe on interventions or policies that were unsuccessful. Third, using qualitative methods to understand prior historical events was challenging, as interviewees often spoke about current experiences rather than discussing historical factors. Additionally, asking about historical events likely introduced some recall bias due to the long study period; participants may not have clearly remembered the inquired events and experiences. Research assistants attempted to mitigate this issue by probing respondents to reflect on longitudinal changes in the immunization program.

These limitations may influence the generalizability of these findings. Further research is required to assess the potential impact of the transferable lessons described above. Similar analyses examining the ICCs of other countries with both low and high immunization coverage may improve the generalizability of this work. Additionally, because we only conducted one round of qualitative data collection, we did not have a chance to collect additional information after our analysis to gain more insight on specific topics that emerged during analysis (e.g., equity, vaccination campaigns, external partnerships).

## 5. Conclusion

Through qualitative research methods, this study sought to further understand the factors that positively affected Zambia's vaccination coverage, including the importance of a strong policy environment to support decision-making and strategic planning. Zambia's status as an exemplar in vaccine delivery was likely supported by the strength, expansion, and prioritization of its national coordination and technical advisory committees. While most countries have functional ICCs, the Zambian ICC demonstrated enhanced strategic planning, decision-making, and lobbying efforts regarding vaccine programming. More research is needed to explore ICC functionality, compare ICCs from different countries, explore costs related to ICC activities, and assess the integral relationship between the ICC and NITAG. Findings from this paper may contribute to the decision-making processes, long-term engagement, membership, and mandates for ICCs in other countries.

## Supporting information

**S1 Checklist. Inclusivity in global research.**
(DOCX)

## Author Contributions

**Conceptualization:** Kyra A. Hester, Walter Orenstein, Matthew C. Freeman, Robert A. Bednarczyk.

**Data curation:** Katie Rodriguez, Bonheur Dounebaine, Anna S. Ellis, William Kilembe.

**Formal analysis:** Zoe Sakas, Katie Rodriguez, Kyra A. Hester, Roopa Darwar, Bonheur Dounebaine, Anna S. Ellis, Simone Rosenblum, Kimberley R. Isett, Walter Orenstein, Matthew C. Freeman, William Kilembe, Robert A. Bednarczyk.

**Funding acquisition:** Matthew C. Freeman, Robert A. Bednarczyk.

**Methodology:** Katie Rodriguez, Kyra A. Hester.

**Project administration:** Kyra A. Hester, Anna S. Ellis.

**Supervision:** Matthew C. Freeman, Robert A. Bednarczyk.

**Writing – original draft:** Zoe Sakas, Katie Rodriguez, Roopa Darwar, Bonheur Dounebaine.

**Writing – review & editing:** Zoe Sakas, Katie Rodriguez, Kyra A. Hester, Roopa Darwar, Bonheur Dounebaine, Anna S. Ellis, Simone Rosenblum, Kimberley R. Isett, Walter Orenstein, Matthew C. Freeman, William Kilembe, Robert A. Bednarczyk.

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
