## [Decision Letter · Decision Letter 0]

11 Oct 2023

PGPH-D-23-01334

The role of Zambia’s expansive Inter-agency Coordinating Committee (ICC) in supporting evidence-based vaccine and health sector programming

Dear Dr. Bednarczyk,

Thank you for submitting your manuscript to PLOS Global Public Health. After careful consideration, we feel that it has merit but does not fully meet PLOS Global Public Health’s publication criteria as it currently stands. Therefore, we invite you to submit a revised version of the manuscript that addresses the points raised during the review process.

We look forward to receiving your revised manuscript.

Kind regards,

Carl Abelardo T. Antonio

Academic Editor

Journal Requirements:

3. Please provide separate figure files in .tif or .eps format only and remove any figures embedded in your manuscript file. Please also ensure all files are under our size limit of 10MB.

4. Some material included in your submission may be copyrighted. According to PLOS’s copyright policy, authors who use figures or other material (e.g., graphics, clipart, maps) from another author or copyright holder must demonstrate or obtain permission to publish this material under the Creative Commons Attribution 4.0 International (CC BY 4.0) License used by PLOS journals. Please closely review the details of PLOS’s copyright requirements here: PLOS Licenses and Copyright. If you need to request permissions from a copyright holder, you may use PLOS's Copyright Content Permission form.

Potential Copyright Issues:

Fig 3: please (a) provide a direct link to the base layer of the map (i.e., the country or region border shape) and ensure this is also included in the figure legend; and (b) provide a link to the terms of use / license information for the base layer image or shapefile. We cannot publish proprietary or copyrighted maps (e.g. Google Maps, Mapquest) and the terms of use for your map base layer must be compatible with our CC-BY 4.0 license. 

"

Additional Editor Comments (if provided):

I concur with the reviewers' recommendations to further enhance reporting of the study methods and findings, and to elaborate on the discussion of the study results

Reviewers' comments:

Reviewer's Responses to Questions

**Comments to the Author**

1. Does this manuscript meet PLOS Global Public Health’s publication criteria? Is the manuscript technically sound, and do the data support the conclusions? The manuscript must describe methodologically and ethically rigorous research with conclusions that are appropriately drawn based on the data presented.

Reviewer #1: Yes

Reviewer #2: Yes

2. Has the statistical analysis been performed appropriately and rigorously?

Reviewer #1: N/A

Reviewer #2: Yes

3. Have the authors made all data underlying the findings in their manuscript fully available (please refer to the Data Availability Statement at the start of the manuscript PDF file)?

Reviewer #1: No

Reviewer #2: Yes

4. Is the manuscript presented in an intelligible fashion and written in standard English?

Reviewer #1: Yes

Reviewer #2: Yes

5. Review Comments to the Author

Reviewer #1: This is an informative manuscript that reflects on the roles of Zambia's interagency coordinating committee and national immunization technical advisory group in shaping national immunization coverage. My comments below include suggestions for improving the methods, findings, and conclusion sections.

METHODS:

1) Although the authors specify that the primary case selection criterion was DTP1/3 coverage, it would help to further clarify why Zambia was chosen as the focus of this case study over the other two exemplars, especially since including one or both of the other countries could have enabled literal replication in the study design.

2) Please consider including the key informant interview protocol and codebook as appendices.

3) In Lines 129-30, the authors mention that they used "constructs from existing implementation science fameworks" to develop their codebook. Please specify which frameworks were used and include citations.

4) How did the research team ensure that interview transcripts were coded consistently? Were they double-coded?

5) Line 134 mentions a review of the literature and historical documents - was this process different from the Gavi document review described immediately after? If so, please provide details of the search strategy and inclusion/exclusion criteria used to carry out these activities.

6) Please consider including a reflexivity statement to orient readers to how the authors' positionalities have shaped the study design, data collection, and analysis activities.

FINDINGS:

1) The authors might wish to include a table with more quotes from the key informant interviews, as this context could further support the reported findings.

2) If possible to include, it would be interesting to read about how donors and external partners (other than Gavi) informed priority-setting, funding, and other ICC/ZITAG activities and decision-making processes.

3) Did the data reveal anything about the ICC/ZITAG's roles in implementing vaccination campaigns?

4) Please elaborate on the ICC/ZITAG's roles in shaping vaccine equity, as this was noted in Gavi's 2018 joint appraisal but not really reflected in the reported findings.

ADDITIONAL FEEDBACK:

I strongly suggest drafting a formal discussion section (separate from the findings and conclusion) to further elaborate on the transferable lessons and limitations of this study. Potential questions to explore in this section include:

- The "so what" - what should the readers of this paper take away from this case study? The transferable lessons are a good start, but more detail is needed regarding who (or what institutions) should act on these lessons. Although data collection concluded prior to the COVID-19 pandemic, it may also be worth reflecting on how the roles and capacities of ICCs/NITAGs should evolve in response to the pandemic or other large-scale health emergencies.

- Related research: The authors should consider moving their findings on ICCs in other African countries here.

- What surprised you about the Zambian context? Were there any unexpected findings?

- Are there any questions your study did not answer? What lines of inquiry should future studies pursue?

Reviewer #2: My comments are below.

1. Page 17 Section 3.1.2. I suggest that the authors consider incorporating a more balanced discussion by including direct quotes and additional insights from key informants who raised concerns regarding the expansion. Their perspectives shed light on the various issues associated with the proposed expansion and would contribute to a comprehensive understanding of the stakeholders' viewpoints.

2. Page 19 Section 3.2. Adjust the citation in Section 3.2 to adhere to the specified guidelines. Please ensure consistency and accuracy.

3. Page 22 Section 3.4. The authors could enhance the depth of Section 3.4, to include the practical application of the transferable lessons in various countries and contexts. Specific examples and contexts should be highlighted, elucidating the factors that contribute to the adaptability and relevance of these lessons in diverse settings.

4. Page 23 Section 4. Section 4 should include an explanation of how the identified challenges may impact the generalizability of the findings. This discussion will provide clarity on the limitations and considerations necessary when extrapolating the results to broader contexts.

5. Page 23 Section 4. Further clarification is needed in Section 4 to elaborate on challenges related to recall bias. The connection between these challenges and the limitations in accessing historical records should be explicitly explained, offering a more thorough understanding of potential biases in the data.

6. Page 24 Section 5. The authors should reiterate and provide a clear link between the research objectives and the methods employed. Furthermore, the authors should consider how the subsequent findings are analyzed in the context of these RQs, demonstrating how the study addresses its overarching research goals.

6. PLOS authors have the option to publish the peer review history of their article (what does this mean?). If published, this will include your full peer review and any attached files.

**Do you want your identity to be public for this peer review?** For information about this choice, including consent withdrawal, please see our Privacy Policy.

Reviewer #1: No

Reviewer #2: No

While revising your submission, please upload your figure files to th

---

## [Decision Letter · Decision Letter 1]

21 Mar 2024

The role of Zambia’s expansive Inter-agency Coordinating Committee (ICC) in supporting evidence-based vaccine and health sector programming

PGPH-D-23-01334R1

Dear Dr. Bednarczyk,

We are pleased to inform you that your manuscript 'The role of Zambia’s expansive Inter-agency Coordinating Committee (ICC) in supporting evidence-based vaccine and health sector programming' has been provisionally accepted for publication in PLOS Global Public Health.

Best regards,

Carl Abelardo T. Antonio

Academic Editor

Thank you for taking time to provide a point-by-point response to the commenta raised by reviewers on your original submission

Reviewer Comments (if any, and for reference):

Reviewer's Responses to Questions

**Comments to the Author**

1. If the authors have adequately addressed your comments raised in a previous round of review and you feel that this manuscript is now acceptable for publication, you may indicate that here to bypass the “Comments to the Author” section, enter your conflict of interest statement in the “Confidential to Editor” section, and submit your "Accept" recommendation.

Reviewer #1: All comments have been addressed

2. Does this manuscript meet PLOS Global Public Health’s publication criteria? Is the manuscript technically sound, and do the data support the conclusions? The manuscript must describe methodologically and ethically rigorous research with conclusions that are appropriately drawn based on the data presented.

Reviewer #1: Yes

3. Has the statistical analysis been performed appropriately and rigorously?

Reviewer #1: N/A

4. Have the authors made all data underlying the findings in their manuscript fully available (please refer to the Data Availability Statement at the start of the manuscript PDF file)?

Reviewer #1: Yes

5. Is the manuscript presented in an intelligible fashion and written in standard English?

Reviewer #1: Yes

6. Review Comments to the Author

Reviewer #1: Congratulations on a well-revised manuscript! This draft has thoroughly addressed all of my questions and comments.

7. PLOS authors have the option to publish the peer review history of their article (what does this mean?). If published, this will include your full peer review and any attached files.

**Do you want your identity to be public for this peer review?** For information about this choice, including consent withdrawal, please see our Privacy Policy.

Reviewer #1: No
